# Automatic visual concept rankings for large multimodal models

## Abstract

Ensuring the reliability of machine learning models in safety-critical domains such as healthcare requires auditing methods that can uncover model shortcomings. While traditional audits range from costly clinical trials to automatic benchmark evaluations, recent advances in automatic interpretability use AI systems to explain other AI models at scale. We introduce an algorithm for identifying salient visual concepts within large multimodal models (LMMs) and demonstrate that leveraging model internals yields more causally relevant insights than black-box approaches. Applying our method to two medical tasks (skin lesion classification and chest radiograph interpretation), we both uncover verifiable conceptual dependencies of LMMs and identify ways in which automatic concept labels may be misleading, highlighting both the promise of automatic interpretability for auditing and the continued importance of expert-in-the-loop oversight.

## 1 Introduction

In safety-critical domains such as healthcare, there is demand for methods to ensure the reliability of machine learning models and identify their potential shortcomings (Kim et al., 2025). Approaches to model auditing comprise a wide spectrum of user effort and difficulty, ranging from costly prospective clinical trials with manual grading of outputs (Han et al., 2024), to automatic benchmarks and evals on publicly available datasets Johri et al. (2025). The research area of AI interpretability aims to assist in the auditing process by uncovering the reasons why models behave the way they do. For example, if developers can identify that a model is reliant on "shortcuts" (DeGrave et al., 2021), they can tweak the model in ways that make it more likely to generalize robustly in deployment (Janizek et al., 2020).

A recent trend within the field of AI interpretability is a set of techniques known as "automatic interpretability," which use AI models to explain other AI models. These approaches have been necessitated by trends like mechanistic interpretability (Nanda, 2023), which frequently involve explaining vast numbers of internal model features. For example, each individual sparse autoencoder (SAE) trained by Paulo et al. (2024) had 131K features, leading to "millions of latents across multiple models, layers, and SAE architectures" requiring explanation. Outside of mechanistic interpretability, understanding even a single layer of a single model may exceed what is feasible for a human auditor if vast numbers of hypotheses need to be tested. Approaches in automatic interpretability make use of vision-language models (VLMs) like CLIP (Radford et al., 2021) to generate concept labels that can be used for probing (Kim et al., 2024; Oikarinen & Weng, 2022), large multimodal models (LMMs) to identify concepts common to the top activating examples for particular features (Paulo et al., 2024; Zhang et al., 2024), or even "agentic" programs where LMMs are equipped with interpretability tools (Shaham et al., 2024; Bricken et al., 2025).

Our goal was to assess the suitability of automatic interpretability for auditing the visual concepts used by large multimodal models (LMMs), with a specific focus on the application area of medical tasks. We propose an algorithm for identifying and ranking the most salient visual concepts used by large multimodal models (**Visual Concept Ranking**, or VCR). We then demonstrate how approaches using model internals (such as ours) can lead to the discovery of more causally-relevant concepts than approaches that treat the models to be audited as a black box. We finally apply our approach to two medical tasks: the classification of skin lesions as benign or malignant, and the identification of normal chest radiographs. We show how our approach is capable of identifying

interesting conceptual dependencies of the model. We also show how some automatically generated concept labels may be misleading, and how despite the promises of human-free auditing, a human-in-the-loop is still highly beneficial.

## 2 OUR VISUAL CONCEPT RANKING (VCR) APPROACH AND RELATED WORK

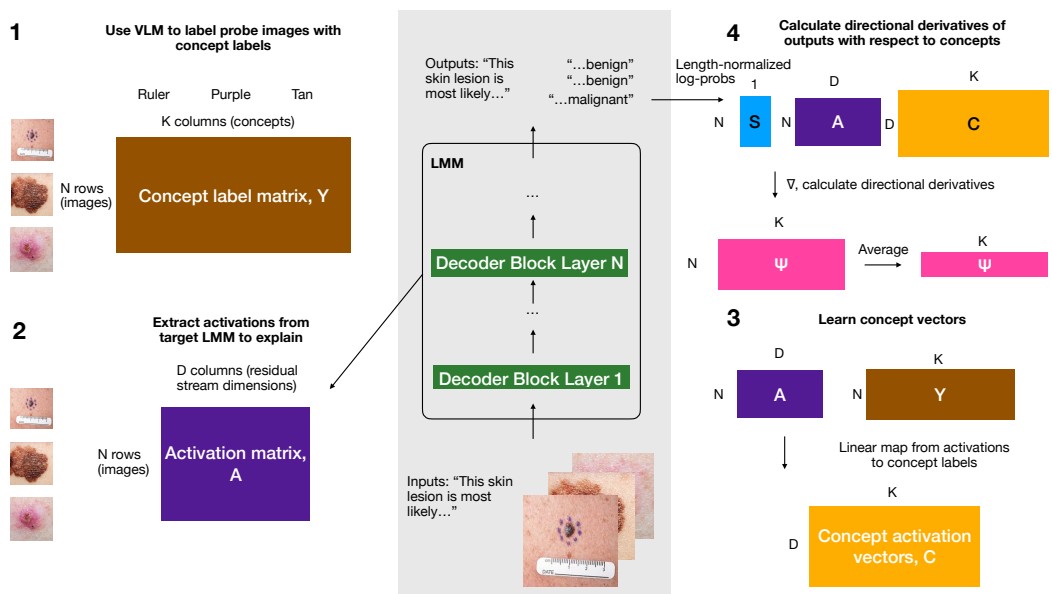

Figure 1: Summary of our method, VCR, to automatically interpret large multimodal models (LMMs) such as OpenFlamingo (center, grey box). **1**, A separate, pretrained vision-language model (e.g. OpenCLIP) is used to label a probe set of $N$ images with continuous values reflecting how much a set of $K$ concepts are represented in each image, generating a concept label matrix $Y \in \mathbb{R}^{N \times K}$. **2**, For each image in the probe set, the corresponding activations from the LMM are extracted into an activation matrix $A \in \mathbb{R}^{N \times D}$, where $D$ is the residual stream dimension. **3**, Concept activation vectors for each concept are calculated by learning linear models to predict each concept label in $Y$ from the activations in $A$. **4**, To calculate the importance of these concept vectors, the directional derivative of the length-normalized log-probs of the target completion of the LMM are taken with respect to each concept, then averaged over all images in the probe set.

Our goal is to identify the most important visual concepts used by a model to complete a task. In this paper, we investigate large multimodal models (LMMs), which take interleaved text and images as input, and produce a probability distribution over natural language tokens as their output. To audit these types of models, we make use of vision-language models (VLMs) such as OpenCLIP (Ilharco et al., 2021), which can take either text or image as input, and output a vector in a joint embedding space.

Our approach is an "automatic" concept labeling approach, meaning that users are not required to manually label probe images with concepts. We assume that the user has a set of images for which they would like to investigate the model's behavior, which do not necessarily require any corresponding class labels: $\mathcal{I} = \{ I_1, I_2, \ldots, I_N \}$. We also assume that the user has a set of concepts they would like to test: $\mathcal{C} = \{ C_1, C_2, \ldots, C_M \}$, where each $C_i$ is a string describing a concept. The concept selection step does not inherently require user expertise, as one can simply use a large set of generic concepts, such as the most common English words in order of frequency (as determined by n-gram frequency analysis of the Google's Trillion Word Corpus (Kaufman, 2013)). However, users with domain expertise may want to specify particular concepts, such as those found in SkinCon (Daneshjou et al., 2022b), a dataset of dermatologist-labeled annotations of important dermatologic concepts in skin lesions. The flexibility to use very large concept sets is a helpful feature, since it enables the discovery of unexpected relationships between concepts and outcomes.

## 2.1 OUTLINE OF VCR ALGORITHM

1. Given an image set $\mathcal{I}$ and a concept set $\mathcal{C}$, we use a pre-trained vision-language model (in all experiments shown here, OpenCLIP) to compute a *concept label* for each image–concept pair (Fig. 1, top left). Each label is the inner product between the image's OpenCLIP embedding and the concept text's embedding, producing a label matrix $Y \in \mathbb{R}^{N \times k}$, where $N = |\mathcal{I}|$ and $k = |\mathcal{C}|$. Intuitively, each entry $Y_{i,j}$ quantifies the extent to which concept $j$ is visually represented in image $I_i$. While we use OpenCLIP in our experiments, we note that any pretrained VLM that has been trained to map images and texts to the same embedding space could be used here.

2. To interpret a target large multimodal model (LMM), such as OpenFlamingo (Awadalla et al., 2023) or LLaVA (Li et al., 2024), we extract hidden activations from a specific layer $\ell$ for each image in the probe set (Fig. 1, bottom left). Following Zou et al. (2023), we use the activations corresponding to the *last token position* from the template text. This yields an activation matrix $A \in \mathbb{R}^{N \times d}$, where each row $a_i = f_\ell(I_i)[-1]$ is the $d$-dimensional residual stream activation vector at layer $\ell$ for the final token for the prompt including an image $I_i$. For each experiment, all components of the prompt (e.g., demonstrations in the ICL setting) are held fixed except for the final image query, and consequently the model's dependence on both the rest of the prompt and the underlying model parameters is omitted from the functional notation.

3. Using the concept labels $Y$ as targets and the activation matrix $A$ as features, we train a linear regression model for each concept $j \in \mathcal{C}$, using $A$ to predict the corresponding column $Y_{.,j}$ (Fig. 1, bottom right). This step is analogous to the *linear probing* stage in RepEng and LG-CAV. The normalized weight vector of the resulting linear model defines the *concept activation vector (CAV)*:

$$v_j = \frac{w_j}{\|w_j\|_2},$$

where $w_j \in \mathbb{R}^d$ is the weight vector learned to predict concept $j$, and $\| \cdot \|_2$ denotes the Euclidean norm. These are collected into a matrix $C \in \mathbb{R}^{d \times k}$, where $k$ again represents the number of concepts.

4. Finally, we assess the *importance of each concept* to the model's output for the task of interest via directional derivatives (Fig. 1, top right). For classification tasks (e.g., predicting whether a skin lesion is "malignant"), we define a *task score* as the length-normalized log-probability of the target class:

$$S_i = \frac{1}{L} \sum_{t=1}^{L} \log \mathbb{P}_f(\text{"malignant"}^{(t)} \mid I_i),$$

where $L$ is the sequence length and the sum is over token positions in the target class label. The *sensitivity of this score to concept $j$* is then defined as:

$$\psi_j = \sum_{i=1}^{N} \langle \nabla_{a_i} S(a_i),\, v_j \rangle$$

where $\nabla_{a_i} S(a_i)$ is the gradient of the score with respect to the activations $a_i$. The outer sum is taken over all images in the probe set in order to produce a single global importance score for each concept.

## 2.2 SIGNIFICANCE TESTING

To assess the statistical significance of each concept's global importance score ($\psi_j$), we repeat our VCR process using resampled versions of the image probing set. For each of the 20,000 concepts tested, we then performed a two-sided one-sample t-test comparing the average of the concepts' global importance scores across the repeated runs against the null hypothesis of a global importance score of zero. To conservatively correct for multiple comparisons across all 20,000 concepts, we applied a Bonferroni correction (Bonferroni, 1936).

## 2.3 RELATED WORK

Previous papers have used VLMs for automatic interpretability. For example, CLIP-Dissect (Oikarinen & Weng, 2022) used CLIP to label the most important concepts for individual neurons within a neural network. Kim et al. (2024) first designed a specially-trained dermatology-specific foundation model that they called MONET, then used MONET to find concepts that correlate with a model's output or a model's loss, an approach they called Model Auditing with MONET (MA-MONET). Our VCR approach is most closely related to the Language-Guided CAVs (LG-CAV) method proposed by Huang et al. (2024), which also leverages VLMs to generate supervision signals for concept activation vectors. Our work differs in two key ways. First, LG-CAV is designed for image classification models trained with supervised learning and fixed class logits, whereas VCR is adapted for large multimodal models (LMMs) that produce probability distributions over natural language tokens (see step 4 of our algorithm outline, where a natural language "task score" is defined). Second, LG-CAV presumes access to labeled examples of concept classes (used in its deviation sample reweighting module and classification loss) while VCR requires no ground-truth concept labels. In this sense, VCR can be viewed as an adaptation of LG-CAV tailored to LMMs and the less-structured data settings in which they are typically audited.

Other approaches to automatic interpretability include Zhang et al. (2024), where authors trained SAEs on the activations of a small LMM (LLaVA-NeXT-8B) to produce disentangled latent features, then used a larger LMM (LLaVA-OV-72B) to automatically interpret those latents by generating natural language descriptions from top-activating examples. Similarly, Paulo et al. (2024) build a pipeline using LLMs to automatically generate natural language explanations for SAE features from other LLMs. Beyond automatic interpretability, there has been substantial research into linear representations of concepts and linear probing of neural networks Belinkov (2022); Park et al. (2023). For example, a recent paper by Rajaram et al. (2025) trained linear probes on residual stream activations of LMMs and demonstrated that image features are represented linearly, become more multimodal in deeper layers, and can be manipulated to control model behavior. Kim et al. (2018) introduced Testing with Concept Activation Vectors (TCAV), a method to quantify how much user-defined, human-interpretable concepts influence a model's predictions by measuring directional derivatives along concept vectors in activation space. Zou et al. (2023) develop RepEng, an approach for testing LLMs' sensitivity to particular textual concepts, and steering their use of those concepts.

## 3 RESULTS

### 3.1 VERIFYING CONCEPT RELIABILITY

A key advantage of our VCR approach is that it directly measures the gradient of the model's outputs with respect to concept representations, providing a principled way to identify concepts that causally influence model behavior. To validate that our gradient-based method successfully identifies causally important concepts, we conducted perturbation experiments using the CheXpert (Irvin et al., 2019) and Diverse Dermatology Images (DDI) (Daneshjou et al., 2022a) datasets. Our experimental setup splits each dataset in half: we use one half as the probe set to extract concept vectors from the residual stream in the last decoder block, and reserve the other half for perturbation testing. For each test image, we first compute the unperturbed length-normalized log-probability for a target completion. We then systematically perturb the activations at the last input token position by adding or subtracting learned concept vectors of varying magnitudes. These perturbations are applied via forward hooks that modify activations during the forward pass, allowing us to measure how the model's predictions change when activations are shifted along specific concept directions.

Figure 2 shows results for both OpenFlamingo-3B-Instruct and OpenFlamingo-4B for the task of malignant versus benign classification of skin lesions using the DDI dataset, tested under both zero-shot and ICL conditions (see Appendix Figs. 17 and 18 for exact prompts used), while Fig. 3 shows the same results but for the task of differentiating normal from abnormal radiographs using the CheXpert dataset (see Appendix Figs. 19 and 20 for prompts). We evaluate several concept selection strategies: the top 20 concepts predicted by VCR to increase the model's length-normalized log-probability of completing with "malignant," the top 20 concepts predicted by VCR to decrease this quantity, and variance-weighted versions of both. The variance weighting addresses the intuition that models may exhibit high sensitivity to concepts that don't vary much within the dataset, which may

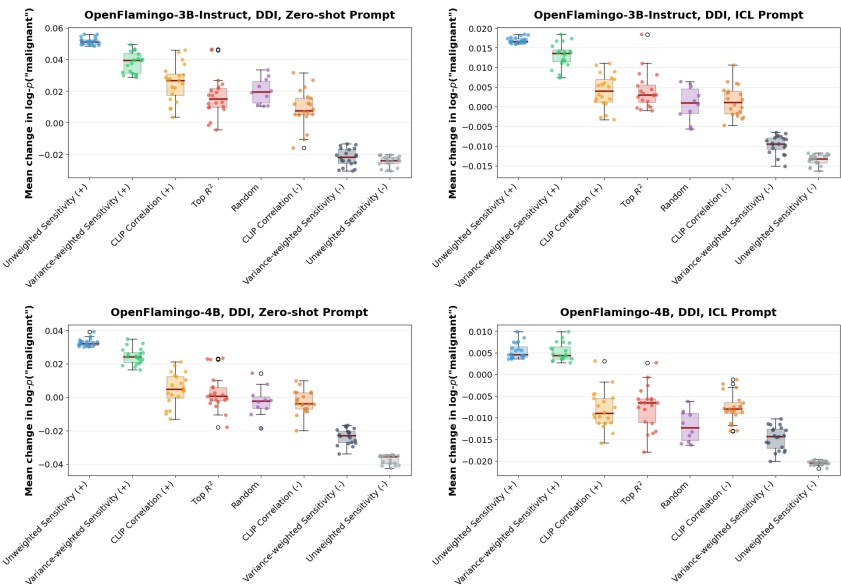

Figure 2: Perturbation validation results for the Diverse Dermatology Images dataset. Each point is the average change in the length-normalized log-probability of the "malignant" completion, averaged over all images in the test set for one of the top 20 concepts identified by that method.

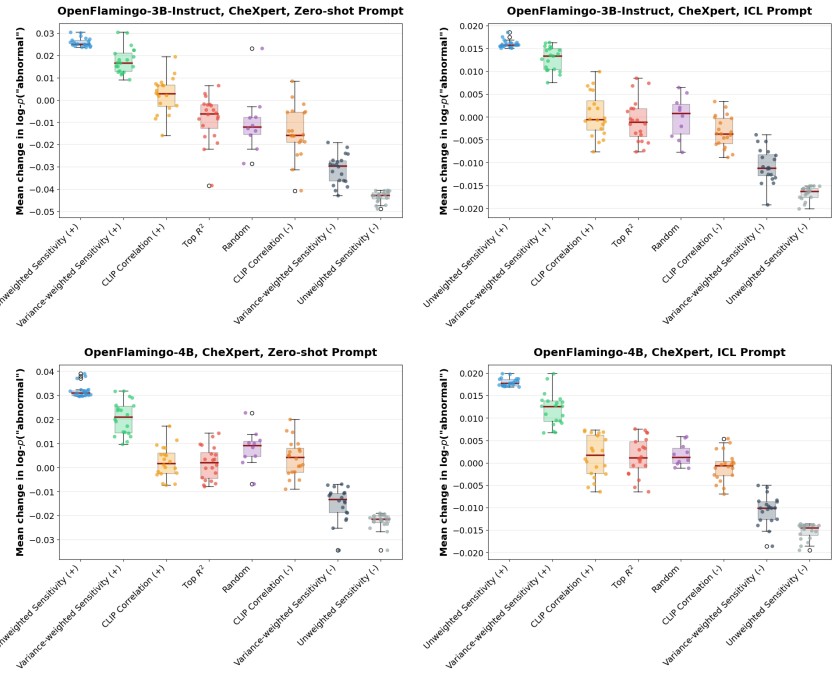

Figure 3: Perturbation validation results for the CheXpert dataset. Each point is the average change in the length-normalized log-probability of the "abnormal" completion, averaged over all images in the test set for one of the top 20 concepts identified by that method.

be less relevant than concepts with more modest sensitivity but a higher variance in representation across the dataset of interest. The specific variance used is the empirical variance in the concept score across the probe set.

For comparison, we include several baselines: correlation-based concept selection (similar to the MA-MONET approach proposed by Kim et al. (2024)), concepts ranked by $R^2$ scores (measuring how well the linear CAV model predicts concept scores from held-out activations), and randomly selected concepts. The $R^2$ baseline quantifies how well each concept is linearly represented within the model layer.

Across models, datasets, and prompt types, our gradient-based scoring consistently identifies concepts that produce larger perturbation effects on model outputs (for both the raw VCR scores and the variance-weighted scores). This validates our core hypothesis that gradient-based concept selection more effectively captures causal relationships between concepts and model behavior compared to correlation-based approaches. The results demonstrate that high correlation with model outputs does not necessarily indicate causal influence, a distinction that becomes crucial for understanding model decision-making processes. These findings underscore the value of leveraging model internals for concept discovery and highlight how gradient-based methods can provide more reliable insights into the causal structure of model representations.

### 3.2 SCALING NUMBER OF CONCEPTS TESTED

We also analyzed the wall-clock time of VCR (see Fig. 4A). Using the 328 train samples from the DDI dataset (half of the dataset) with the zero-shot prompt, we found that it took approximately 60 to 90 seconds on a single NVIDIA GeForce RTX 4090 GPU to generate between 500 and 20K concept explanations for a single layer. Importantly, we see that adding additional concepts did not significantly impact the overall time required, adding *only 8 seconds* to generate 20K vs 500 concept explanations (Fig. 4B). This is because the process of generating VLM embeddings and training CAVs was relatively small compared to fixed costs like loading the LMM onto the GPU, the forward passes required to extract log-probs and activations, and the backward passes required to calculate directional derivatives. In the Appendix, we also analyze the impact of changing the probe set size, which has a much larger impact on the total time required. This effect is linear in the number of images in the probe set, as this increases the number of forward passes and backward passes of the model required. We highlight here that our approach remains so computational feasible that we are also able to use *an order of magnitude* larger probe set sizes than those described in prior work such as Kim et al. (2018), where they demonstrated effective CAV training with approximately 30 example images. Our probe set sizes were similar to prior automated approaches like LG-CAV (Huang et al., 2024).

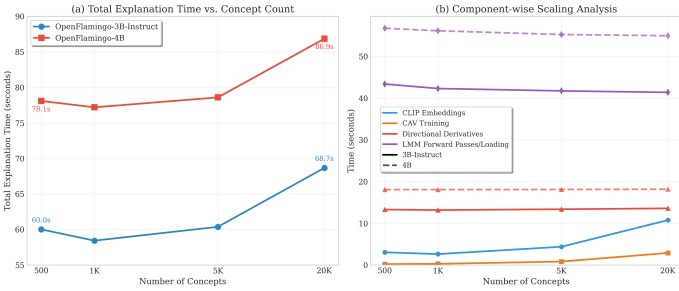

Figure 4: Wall-clock time to generate explanations for the last layer of two OpenFlamingo models, as the number of concept explanations generated is varied.

### 3.3 VISUALIZING THE EVOLUTION OF CONCEPTS ACROSS LAYERS

Given that we were able to generate explanations efficiently, we were consequently able to run a variety of model-wide analyses, looking at how concepts changed across layers in the residual stream.

For example, for our final interpretations in the next section, we wanted to focus on the residual stream following the last decoder block, as this is the final representation developed by the model prior to producing a natural language output. However, prior literature by Nicolson et al. (2024)

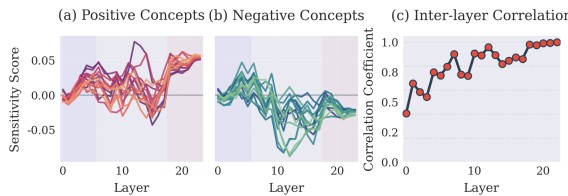

Figure 5: Evolution of the top 15 most influential (positive and negative) concepts across the layers of the OpenFlamingo-3B-Instruct model for a zero-shot dermatology prompt on the DDI dataset.

warns that concept representations can be inconsistent across layers, potentially leading to unreliable interpretations. We therefore conducted a layer-wise consistency analysis for our top VCR-identified concepts. For each of the most important 15 positive and negative concepts, we measure the evolution of that concept's sensitivity score ($\psi$) across all layers in the model. In Fig. 5, we see that while there is variability in concept sensitivity across initial layers, all of the concepts show a high degree of stability by the final layers, with a nearly perfect average correlation between concept sensitivity across layers in the last several layers (Fig 5C). We repeat this analysis for several other models and prompts in the Appendix.

We can also test hypotheses related to how the linear representation quality and model sensitivity of certain concepts are impacted by the prompt used for a task. Using images from DDI paired with either a dermatological prompt (where the model is tasked with differentiating benign from malignant skin lesions), and a non-dermatological prompt (where the model is tasked with predicting which country a picture was taken in), we examine how both the the VCR-determined model sensitivity and the linear representation quality ($R^2$ score) change for skin-specific concepts (from SkinCon) and all concepts. We see that as we progress to later layers in the model, if the model is prompted with a dermatological task, not only is the output more sensitive to skin-related visual concepts compared to all visual concepts (Fig. 6), but the linear representation quality of the skin-related visual concepts is relatively higher than average visual concepts (Fig. 7). When the model is prompted with a non-dermatological task, both the sensitivity and the linear representation quality for the dermatology concepts are lower than average. In the Appendix, we conduct the same analysis for the OpenFlamingo-4B model, and find that while the same trend holds for linear representation quality (Appendix Fig. 15), there is no significant difference in the model's relative sensitivity to skin-related concepts (Appendix Fig. 14).

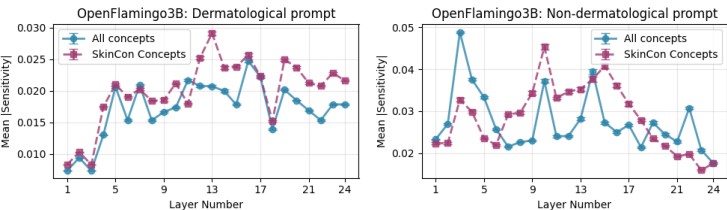

Figure 6: Evolution of model sensitivity to visual concepts (VCR $\psi$ score) across layers of the OpenFlamingo-3B-Instruct model.

### 3.4 INSPECTION OF TOP VISUAL CONCEPTS

After systematically sanity checking the reliability of our approach with the prior analyses, we used VCR to interpret different large multimodal models (LMMs) for realistic medical tasks. All results shown in the main text of the paper look at the top concepts from the residual stream following the last decoder block of the OpenFlamingo-3B-Instruct model using zero-shot prompts (see Appendix Fig 17), where the model is prompted to either classify a skin lesion as malignant or benign based on an attached image or classify a radiograph as normal or abnormal.

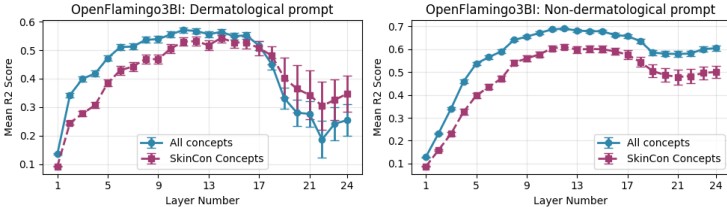

Figure 7: Evolution of linear representation quality of visual concepts across layers of the OpenFlamingo-3B-Instruct model, as determined by $R^2$ score of CAVs predicting concept labels ($Y$) from held out samples' activations.

When we examine the top concepts for the dermatology task, we find that many are apparently related to the location on the patient's body where the lesion is found. For example, a visual concept labeled "hearing" (which activates in response to images of ears) significantly decreases the likelihood of calling a lesion malignant (see Fig. 8), while a visual concept labeled "abdominal" (see Appendix Fig. 16) significantly increases the model's likelihood of calling a lesion malignant. This is an interesting finding for several reasons. First, these are medically plausible visual concepts to use for the classification task; there are a variety of well-described associations between bodily regions and different malignancies, such as an increased likelihood of cutaneous T cell lymphoma to manifest on "doubly-covered" areas of the skin (normally covered by two layers of clothing) (Semaan et al., 2021), or a predilection for melanoma to develop on the head/neck and trunk (Cho et al., 2005). Second, identification of these concepts highlights the advantage of using a very large concept set, as compared to having experts manually define a small set of concepts that will be tested. Had we just used the expert-defined dermatology concepts in SkinCon as probes, for example, we would not have identified that the models tended to look at these bodily regions.

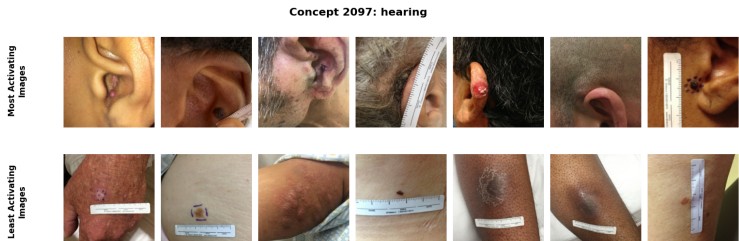

Figure 8: Most (top row) and least (bottom row) activating images for OpenFlamingo-3B-Instruct's "hearing" concept for the task of differentiating malignant from benign skin lesions. This was one of the top visual concepts decreasing the model's likelihood of classifying a lesion as malignant.

We also find some shortcomings of the automatic labeling approach. For example, one of the visual concepts identified by VCR as important for the model is a concept labeled as "Purpura/Petechiae." This refers to discolored spots of skin that result from leaking blood vessels, and is certainly plausible as the sort of visual dermatologic concept that a model might rely upon. However, when we examine the top activating images for the concept (see Fig. 9), we see that this concept not only activates for petechial lesions, but also activates for the purple ink markings used by dermatologists to outline a suspicious skin lesion for biopsy. This is a classic "shortcut," where the model looks at features that are causally downstream of an abnormality (i.e. markings that a physician has applied to a patient in preparation for a biopsy, indicating that the physician thought the lesion was suspicious), rather than looking for the visual features of the abnormality itself. This highlights the importance of a human or expert-in-the-loop. Even if a VLM is used to make a first pass at annotating the concepts that will be tested for associations, a human expert is still needed at this point to ensure complete accuracy. We also want to emphasize that this does not mean our method is not useful – even if it requires human intervention to arrive at the correct final labeling, it is still capable of identifying semantically-meaningful concepts that are causally related to the model's output for a task of interest, sorted in terms of their magnitude of impact on the model's output.

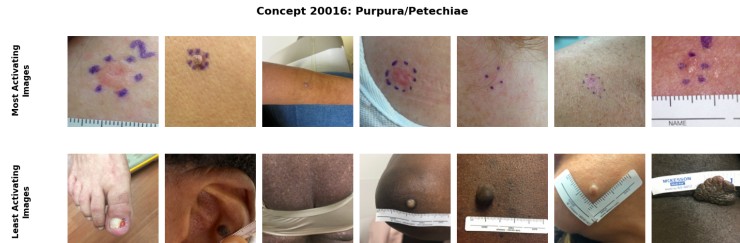

Figure 9: Most (top row) and least (bottom row) activating images for OpenFlamingo-3B-Instruct's "Purpura/Petechiae" concept for the task of differentiating malignant from benign skin lesions. This was one of the top visual concepts increasing the model's likelihood of classifying a lesion as malignant.

Our approach works outside of dermatology on a radiology task as well. VCR identifies that the OpenFlamingo-3B-Instruct model tends to look at a visual concept labeled as "wiring" which activates for medical support devices like EKG leads or pacemakers (Fig. 10) to increase it's likelihood of calling a radiograph abnormal, which is another classic example of a "shortcut."

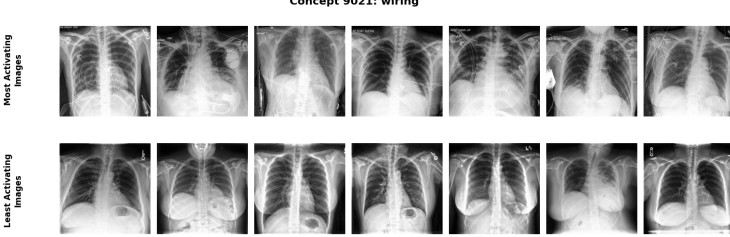

Figure 10: Most (top row) and least (bottom row) activating images for OpenFlamingo-3B-Instruct's "wiring concept" for the task of differentiating normal from abnormal chest radiographs. This was one of the top visual concepts increasing the model's likelihood of classifying a lesion as abnormal.

## 4 DISCUSSION, LIMITATIONS, AND FUTURE WORK

We introduce VCR as a principled method for identifying causally relevant visual concepts in LMMs through gradient-based analysis, demonstrating clear advantages over correlation-based approaches across medical tasks. Our work reveals both methodological limitations and promising directions for advancing automatic interpretability.

**Automatic concept labeling:** Our most significant finding is that automatic concept labeling can produce misleading results without human oversight. The "Purpura/Petechiae" example, where an apparently accurate concept label actually captured purple ink markings rather than skin pathology, illustrates a fundamental challenge with relying on AI for concept labeling – concept labels may only be as accurate as the VLM used to generate them. However, as progress continues and as models continue to improve, one advantage of our approach is that its modularity/flexibility will allows us to replace OpenCLIP with newer, more accurate VLMs. Hence, we believe that an important research focus will be *developing more accurate foundation models*, and this will naturally lead to further improvements in interpretability.

**Further applications:** While we look at several LMMs (OpenFlamingo-3B-Instruct and OpenFlamingo-4B) across several different realistic medical datasets (DDI and CheXpert) and several different prompt types, there is obviously a huge range of potential applications not explored in depth in this paper. We briefly investigate a few extensions in the Appendix, including a larger 9B parameter LMM adapted specifically for medical data, and including experiments with natural images rather than medical images. However, we believe this merely scratches the surface of future application/domain-specific work with our method.

## 5 ETHICS STATEMENT

This work focuses on interpretability methods for medical AI systems, where improved transparency generally should benefit patient safety. However, the techniques could theoretically be misused to identify model vulnerabilities or craft adversarial inputs. The reliance on particular vision-language models for concept labeling may also perpetuate inaccuracies present in these models' training data, as demonstrated by our "Purpura/Petechiae" example where automatic labels were misleading. While we focus on "automatic" interpretability, we believe our paper also demonstrates how human oversight remains essential for validating concept labels, particularly in safety-critical domains.

## 6 REPRODUCIBILITY STATEMENT

In order to aid in reproducibility, we include an anonymized repository of code containing our VCR module in the Supplementary Material. This has instructions to install and download the necessary datasets and software dependencies, as well as example scripts to generate explanations for both CheXpert and DDI tasks. In the final published version of the paper, we will make our easy-to-use open source library public. This will not only improve the reproducibility of the results within this paper, but will also allow others to apply our tool to their own research questions and applications.

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

# A APPENDIX

## A.1 SUPPLEMENTAL FIGURES FROM THE MAIN PAPER

Supplemental figures from the main text include Appendix Figs. 11 through 13, which show the stability of top concepts across the residual stream in the final layers of the models; Appendix Figs. 14 and 15, which show how model sensitivity and linear representation quality evolve across layers for the 4B OpenFlamingo model; and Appendix Fig. 16, which shows the most and least activating images for the "abdominal" visual concept for OpenFlamingo-3B-Instruct on the DDI dataset.

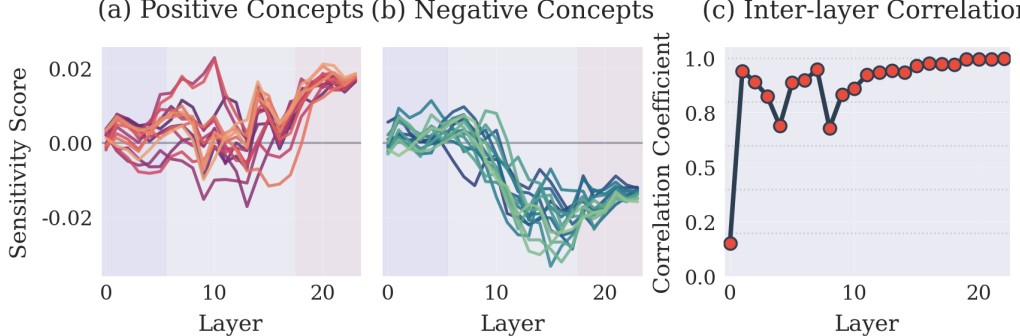

Figure 11: Evolution of the top 15 most influential (positive and negative) concepts across the layers of the OpenFlamingo-3B-Instruct model for the ICL dermatology prompt on the DDI dataset.

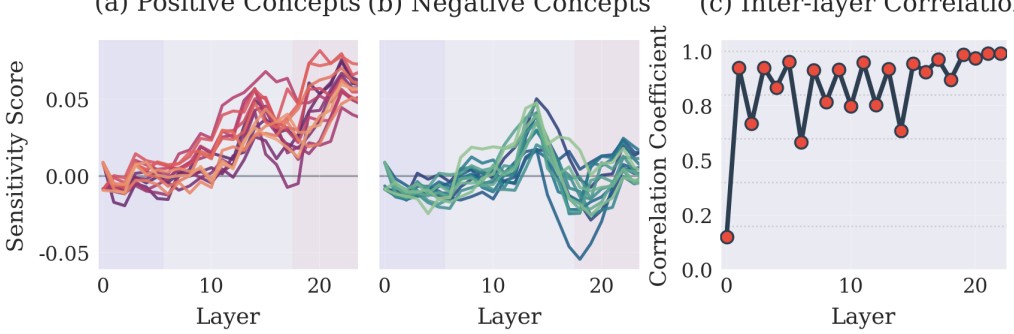

Figure 12: Evolution of the top 15 most influential (positive and negative) concepts across the layers of the OpenFlamingo-4B model for the zero-shot dermatology prompt on the DDI dataset.

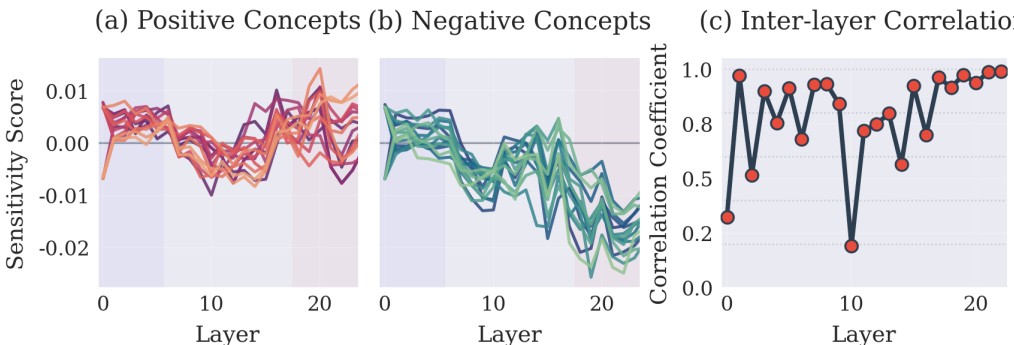

Figure 13: Evolution of the top 15 most influential (positive and negative) concepts across the layers of the OpenFlamingo-4B model for the ICL dermatology prompt on the DDI dataset.

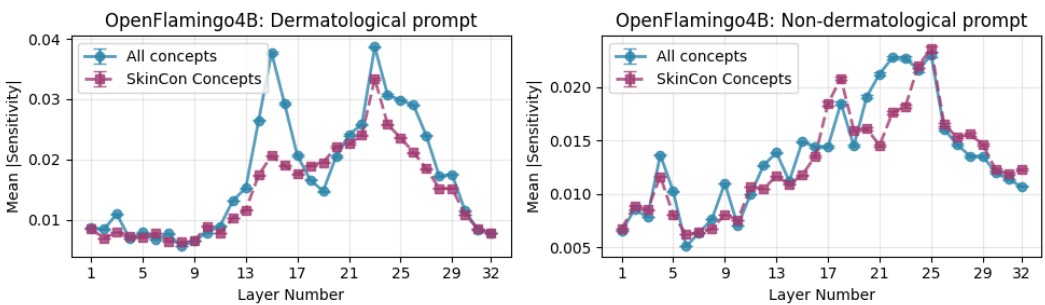

Figure 14: Evolution of model sensitivity to visual concepts (VCR concept sensitivity scores, $\psi$) across layers of the OpenFlamingo-4B model.

## A.2 TASK PROMPTS

Appendix Figures 17 through 20 contain the exact prompts used for the DDI and CheXpert experiments.

## A.3 FURTHER EXPERIMENTAL DETAILS ON COMPUTATIONAL COST EXPERIMENTS

We conducted two timing experiments to assess the computational scalability of our concept-based explanation approach using OpenFlamingo-3B-Instruct and OpenFlamingo-4B models. All experiments were performed on a single NVIDIA 4090 RTX, using identical hardware and software configurations to ensure fair comparison.

The timing analysis focused on five key computational components: (1) model loading, (2) CLIP embedding computation, (3) concept model training, (4) directional derivatives calculation, and (5) other processing steps including activation collection and concept weight computation. Each experiment was run with a fixed random seed to ensure reproducible data splits and timing measurements.

To evaluate how explanation time scales with concept vocabulary size, we varied the number of concepts while holding other parameters constant. We tested four concept vocabulary sizes: 500, 1,000, 5,000, and 20,000 concepts. The probe set was constructed by randomly sampling training images from the DDI dataset, with a fixed train/test split ratio of 0.5 and consistent data preprocessing across all concept vocabulary sizes. We used layer 23 for OpenFlamingo-3B-Instruct and layer 31 for OpenFlamingo-4B, representing the final decoder layers in each model architecture.

To assess computational scaling with respect to training data size, we varied the probe set size while maintaining a fixed concept vocabulary of 1,000 concepts. We tested three probe set sizes: 50, 100, and 200 training samples. For each probe set size, we randomly sampled the specified number

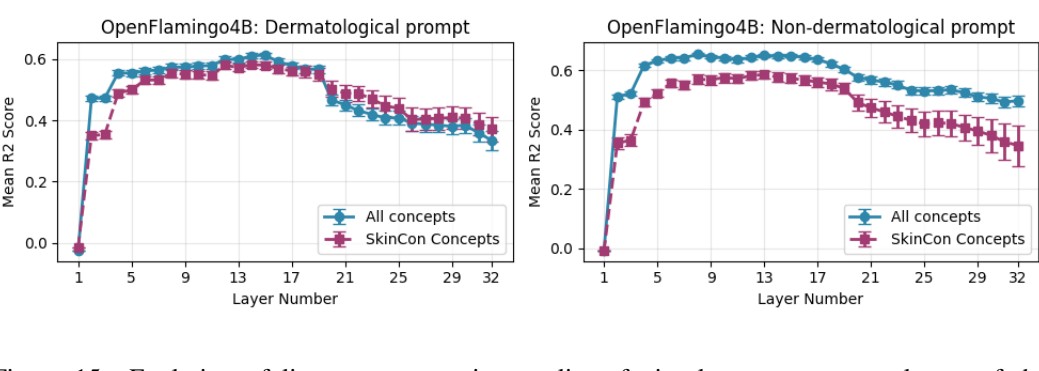

Figure 15: Evolution of linear representation quality of visual concepts across layers of the OpenFlamingo-4B model.

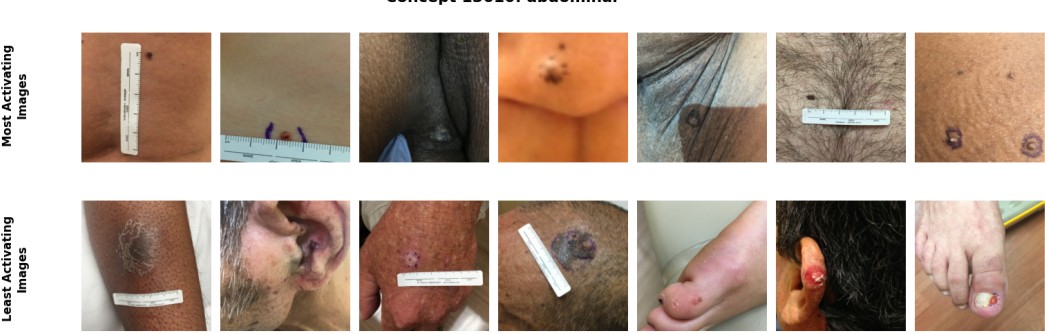

Figure 16: Most (top row) and least (bottom row) activating images for OpenFlamingo-3B-Instruct's "abdominal" concept for the task of differentiating malignant from benign skin lesions. This was one of the top visual concepts increasing the model's likelihood of classifying a lesion as malignant.

of training images from the DDI dataset, with remaining samples allocated to the test set. This design allowed us to isolate the computational impact of increasing the number of images for which activations must be collected and gradients computed, while controlling for concept vocabulary complexity. The same model layers and processing pipeline were used as in Experiment 1.

Each experiment measured wall-clock time for individual computational components using Python's time module. We recorded the duration of each major processing step and computed total explanation time as the sum of all components. To account for potential GPU memory and caching effects, we performed cleanup operations between different parameter settings, including removal of temporary files and explicit garbage collection. The timing measurements capture the end-to-end computational cost of generating explanations for a single model layer, including all preprocessing, model inference, and post-processing steps required by our approach.

## A.4 APPLYING VCR TO A LARGER MODEL

As mentioned in our Discussion section, the applications demonstrated in the main text likely merely scratch the surface of potential application of our VCR approach. To demonstrate that our approach can be applied to other models, data, and task types, we generated VCR explanations for a larger, 9B parameter LMM called MedFlamingo. We focused on the self-attention modules within the decoder layers. We used natural images from ImageNette to test the model, specifically selecting two distinct object classes: Fish images (Class n01440764 from Imagenette) and Golf ball images (Class n03445777 from Imagenette). The dataset was split 50/50 into training and test sets using a fixed random seed (1017) to ensure reproducibility. From the training set, we reserved 2% of examples as demonstration images for in-context learning. We employed 3,000 diverse concept terms from a broad vocabulary file. We designed a few-shot learning template to elicit binary classification

**DDI Dataset Prompt Template Components:**

**Zero-shot Base Prompt:**

```
"Based on the image, this lesion is
benign.<|endofchunk|>Based on the image,
this lesion is malignant.<|endofchunk|>"
```

**Query Template:**

```
"<image>Based on the image, this lesion is"
```

**Complete Zero-shot Example:**

```
"Based on the image, this lesion is
benign.<|endofchunk|>Based on the image, this
lesion is malignant.<|endofchunk|><image>Based on
the image, this lesion is"
```

Figure 17: Zero-shot prompt template used for the DDI (dermatology) dataset. Following the original Flamingo paper, the base prompt provides example plain text completions (without images), followed by the query image and incomplete prompt for classification.

**DDI Dataset In-Context Learning Components:**

**Base Prompt:**

```
"Based on the image, this lesion is
benign.<|endofchunk|>Based on the image,
this lesion is malignant.<|endofchunk|>"
```

**Demonstration Template:**

```
"<image>Based on the image, this lesion is
{label}.<|endofchunk|>"
```

**Query Template:**

```
"<image>Based on the image, this lesion is"
```

**Complete ICL Example:**

```
"Based on the image, this lesion is
benign.<|endofchunk|>Based on the image, this
lesion is malignant.<|endofchunk|><image>Based
on the image, this lesion is
benign.<|endofchunk|><image>Based
on the image, this lesion is
malignant.<|endofchunk|><image>Based on the
image, this lesion is"
```

Figure 18: In-context learning prompt template for the DDI dataset. The base prompt is followed by demonstration examples with actual images and labels, then the query image for classification.

behavior (see Appendix Fig. 22). The resulting top concepts (see Appendix Table 1) demonstrate that: (1) the model's fish concepts are semantically coherent (biological, aquatic terms), (2) Golf ball concepts relate to sports equipment and visual patterns (see Appendix Fig. 23, which shows that a dotted visual pattern is important, and Appendix Fig. 24, which shows that the visual presence of logos in the image leads a model to predict an image is a golf ball). The model successfully learns to distinguish domain-specific features at the attention layer level, not just in the residual stream.

A.5 LLM USAGE

Per this year's ICLR policy on disclosing LLM usage in the creation of papers, both ChatGPT and Claude were used to generate feedback on our writing, including suggestions for clarification and

---

**CheXpert Dataset Prompt Template Components:**

**Zero-shot Base Prompt:**

```
"Based on the radiograph, this
study is normal.<|endofchunk|>Based
on the radiograph, this study is
abnormal.<|endofchunk|>"
```

**Query Template:**

```
"<image>Based on the radiograph, this study
is"
```

**Complete Zero-shot Example:**

```
"Based on the radiograph, this study is
normal.<|endofchunk|>Based on the radiograph,
this study is abnormal.<|endofchunk|><image>Based
on the radiograph, this study is"
```

---

Figure 19: Zero-shot prompt template used for the CheXpert (chest X-ray) dataset. Following the original Flamingo paper, the base prompt provides example plain text completions (without images), followed by the query image and incomplete prompt for classification.

---

**CheXpert Dataset In-Context Learning Components:**

**Base Prompt:**

```
"Based on the radiograph, this
study is normal.<|endofchunk|>Based
on the radiograph, this study is
abnormal.<|endofchunk|>"
```

**Demonstration Template:**

```
"<image>Based on the radiograph, this study
is {label}.<|endofchunk|>"
```

**Query Template:**

```
"<image>Based on the radiograph, this study
is"
```

**Complete ICL Example:**

```
"Based on the radiograph, this study is
normal.<|endofchunk|>Based on the radiograph,
this study is abnormal.<|endofchunk|><image>Based
on the radiograph, this study is
normal.<|endofchunk|><image>Based
on the radiograph, this study is
abnormal.<|endofchunk|><image>Based on the
radiograph, this study is"
```

---

Figure 20: In-context learning prompt template for the CheXpert dataset. The base prompt is followed by demonstration examples with actual images and labels, then the query image for classification.

potential improvements. All writing in its final form in the submitted text was written by one of the human authors. Additionally, FutureHouse's Platform (specifically the Owl Precedent Search tool) was used for researching relevant prior work, to help make sure we were referencing all relevant related scholarship.

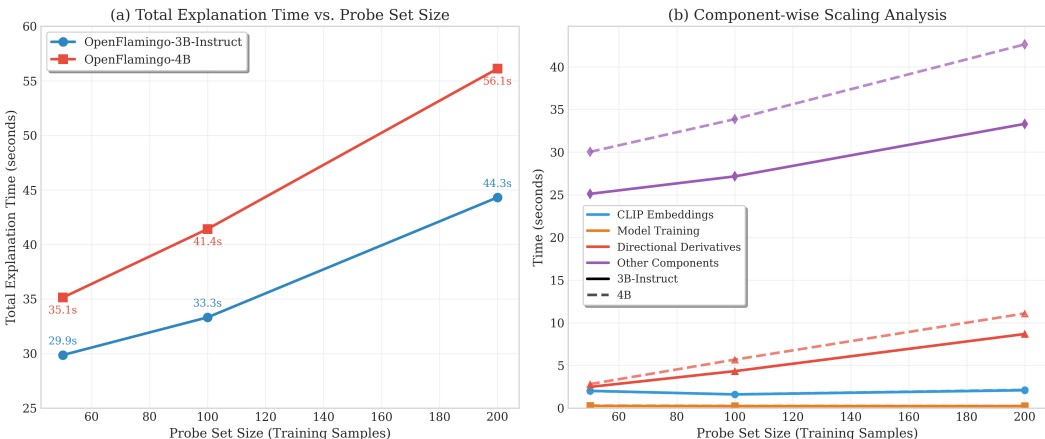

Figure 21: Wall-clock time to generate explanations for the last layer of two OpenFlamingo models, as the number of images in the probe set is varied.

---

**Imagenette Prompt Template Components:**

**Base Prompt:**

```
"This is an image of a fish.<|endofchunk|>This
is an image of a golf ball.<|endofchunk|>"
```

**Demonstration Template:**

```
"<image>This is an image of a
{label}.<|endofchunk|>"
```

**Query Template:**

```
"<image>This is an image of a"
```

**Complete Example:**

```
"This is an image of a fish.<|endofchunk|>This is
an image of a golf ball.<|endofchunk|><image>This
is an image of a fish.<|endofchunk|><image>This
is an image of a golf ball.<|endofchunk|><image>This
is an image of a"
```

---

Figure 22: Prompt template structure for few-shot Imagenette classification. The template combines base instruction, demonstration examples, and query formatting to elicit consistent classification behavior from the MedFlamingo model.

Table 1: Top 25 visual concepts for Fish vs Golf Ball Classification in MedFlamingo's 25th decoder layer self-attention module.

| Fish Concepts | | Golf Ball Concepts | |
|---|---|---|---|
| aquaculture | fisheries | putting | logos |
| suriname | fish | tights | dotted |
| carp | freshwater | incorporates | pointing |
| mutant | fishing | plaid | stripes |
| bass | peruvian | visor | affiliations |
| female | specimen | pointers | overall |
| biodiversity | fishery | iu | acura |
| unit | discarded | pantera | lettering |
| marrow | futuna | docs | paige |
| syndicate | meal | scottsdale | pines |
| nunavut | pike | memberships | belts |
| sphere | guyana | tapestry | rings |
| syrian | | classicvacations | |

**Concept 19400: dotted**

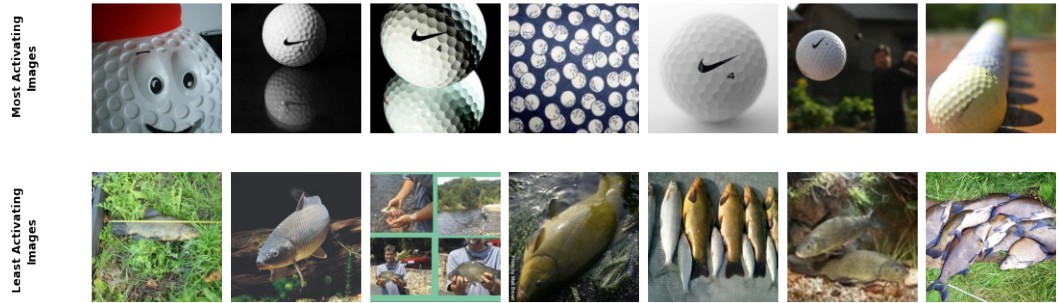

Figure 23: Most (top row) and least (bottom row) activating images for MedFlamingo's "dotted" concept for the task of differentiating fish from golf balls using images from the Imagenette dataset. This was one of the top visual concepts increasing the model's likelihood of classifying an image as containing a golf ball.

**Concept 3920: logos**

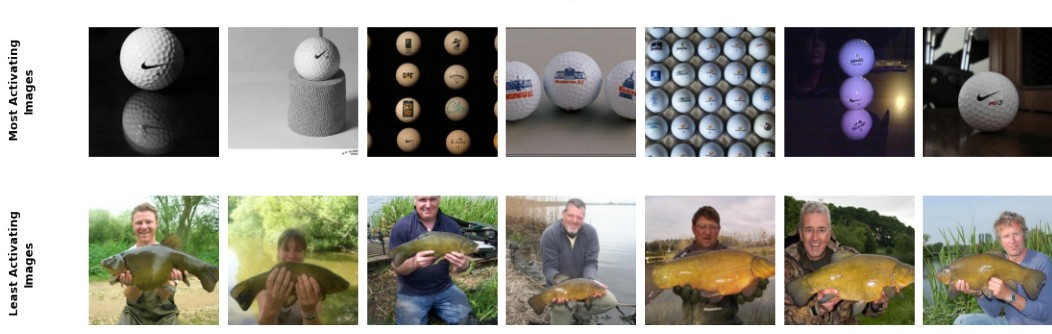

Figure 24: Most (top row) and least (bottom row) activating images for MedFlamingo's "logos" concept for the task of differentiating fish from golf balls using images from the Imagenette dataset. This was one of the top visual concepts increasing the model's likelihood of classifying an image as containing a golf ball.

