# OpenReview forum: "Automatic visual concept rankings for large multimodal models"
_ICLR.cc/2026/Conference — Submitted to ICLR 2026_

### Official Review · Reviewer_Ey7F · 2025-10-27

**Soundness:** 3
**Presentation:** 3
**Contribution:** 2
**Rating:** 4
**Confidence:** 4

**Summary:**

This paper introduces Visual Concept Ranking (VCR), an interpretability method designed to explain the output of large multimodal models (LMMs). The core idea is to identify concepts that are causally relevant to the model's output. The method first learns Concept Activation Vectors (CAVs) by probing an LMM's internal activations and then ranks these concepts by calculating the directional derivative of the model's output log-probabilities with respect to these CAVs. The authors demonstrate their method in medical tasks

**Strengths:**

- The paper's primary strength lies in its principled approach to identifying causal concept influence. By using gradient-based directional derivatives, it provides a more robust measure of concept importance.
- The VCR algorithm is clearly explained in four distinct steps (Fig 1) and appears technically sound. The scalability analysis (Fig 4) is also a useful addition.

**Weaknesses:**

- Although mentioned in the article, the key "shortcut" finding (Fig. 9) was not identified by the algorithm itself, but required manual human inspection of the surfaced images. This seems to severely contradict the author's claim of "automatic interpretability."
- Since there are so many existing methodologies related to concept bottleneck, it seems that there should be more comparisons with existing methodologies.

**Questions:**

- It would be helpful to compare your research with Kim et al [1]. This seems essential, as it's similar to a prior study that used llm and clip to automatically generate concepts.
- It would be better if we could see the quality evaluated directly by experts.


[1] Kim, Injae, et al. "Concept bottleneck with visual concept filtering for explainable medical image classification." International Conference on Medical Image Computing and Computer-Assisted Intervention. Cham: Springer Nature Switzerland, 2023.

---

> ### Author Response · Authors · 2025-11-30
> **Thanks for your feedback**
>
> Thanks for the feedback — to respond point-by-point to your questions:
>
> 1. It would be helpful to compare your research with Kim et al [1]. This seems essential, as it's similar to a prior study that used llm and clip to automatically generate concepts.
>
> While the study by Kim is very interesting and useful, it is a different application than our method. The Kim paper uses CLIP to generate concepts labels that are then used in an intermediate representation of the model. Our approach instead evaluates models that have already been trained, and is not a method to train new models.
>
> 2. It would be better if we could see the quality evaluated directly by experts.
>
> This is an interesting idea as well. We could certainly include a figure showing the proportion of concepts in some sample that appear to be “correctly” labeled.

---

### Official Review · Reviewer_T1q8 · 2025-10-31

**Soundness:** 2
**Presentation:** 3
**Contribution:** 3
**Rating:** 4
**Confidence:** 3

**Summary:**

The paper proposes VCR, a method for automatically interpreting large multimodal models by identifying which visual concepts actually drive their predictions. It uses a vision-language model to label image concepts, maps those to the model's internal activations, and measures how changes in each concept affect the model's output. The authors tested it on medical image datasets to demonstrate its effectiveness, though it still depends on the quality of automated labelling and requires human oversight for final interpretation.

**Strengths:**

1) The work presents a gradient-based concept-activation analysis for LMMs, extending LG-CAV.
2) It proposes a label-free interpretability pipeline using OpenCLIP for automated concept generation, improving scalability for large concept sets.
3) The work provides a scalable, generalisable framework for concept-level interpretability in multimodal models.
4) The method is mostly rigorous and reproducible.
5) The paper is well-written and logically structured; it is reasonably accessible.
6) The testing demonstrates practical value on real-world medical datasets, offering insight into model reasoning and shortcut behaviours in safety-critical domains.

**Weaknesses:**

1) Lack of stronger baselines: While the work mentions using methods such as MA-MONET, it is unclear exactly what that entails. Second, the work explicitly states that it extends Language-Guided CAVs (LG-CAV); therefore, it would be more convincing to include LG-CAV as a baseline as well.

2) Lack of understanding of the model size effect: While the work compares 3B and 4B models, we do not know how well the model will work with larger models. The comparison between the 3B and 4B OpenFlamingo models provides minimal insight into scaling behaviour. It remains unclear how the method performs on larger or different LLMs.

3) Human-in-the-loop: While the authors acknowledge the value of expert oversight to mitigate labelling errors, the paper lacks specifics on when and how human review would be incorporated, what criteria experts would apply, or how their input might quantitatively improve results. Clearer workflow definitions and evaluation of inter-rater reliability would strengthen this argument.

4) Lack of investigation of components: The framework relies exclusively on OpenCLIP for automatic concept labelling, but the authors do not explain why it was selected or whether they tested other vision-language models.

5) Causal inference is claimed but not fully established: This phrase kind of weakens the claim: "it's likelihood of calling a radiograph abnormal, which is another classic example of a 'shortcut." The statement implicitly acknowledges that the findings may reflect internal correlations rather than true causal reasoning. The method remains valuable for diagnostic interpretability, but causal claims should be moderated/toned down.

- Minor comment: There is no need to define LMMs multiple times.

**Questions:**

1) Why do you think the t-test is appropriate for the tests you have done? Could you just make the choice briefly in the paper/appendix?
2) Why were the significance tests not discussed in the paper?
3) Are there any assumptions being made about the correlation of the concepts? If so, please list these alternatives and briefly discuss their implications.
4) What were the results like when tested with < 500 concepts, e.g., 20, 30, 50, 100, etc? Was there a specific reason for starting at 500? Please include the justification in the paper.
5) What were the specific reasons for just testing 3B and 4B models? Are there plans to test with other LLMs?
6) Could you elaborate exactly how human oversight would solve the interpretability problem you are trying to address in the paper?
7) The work uses OpenCLIP. Did the authors explore other options? Why/why not?

---

> ### Author Response · Authors · 2025-11-30
> **Thanks for your feedback!**
>
> 1. Why do you think the t-test is appropriate for the tests you have done? Could you just make the choice briefly in the paper/appendix?
>
> We chose t-tests simply to compare the differences in means between two populations, but can certainly update the results to also look at rank-based testing like Wilcoxon tests.
>
> 2. Why were the significance tests not discussed in the paper?
>
> We discussed how the significance tests were performed in the methods section, but left out the full enumeration of significant concepts for the appendix or supplemental data, as we felt this was less interesting compared to how much space it took up in the paper. We can include an abridged version of tables in the main text.
>
> 2. Are there any assumptions being made about the correlation of the concepts? If so, please list these alternatives and briefly discuss their implications.
>
> We didn’t make any explicit assumptions about the correlation between concepts. Bonferroni correction remains valid even if the tests are correlated, but will of course be less powerful if that is the case. As the concepts become more correlated, this correction will become more conservative (which we believe should be desirable, to avoid false positives). We can add this discussion to the text.
>
> 3. What were the results like when tested with < 500 concepts, e.g., 20, 30, 50, 100, etc? Was there a specific reason for starting at 500? Please include the justification in the paper.
>
> The different number of concepts tested were just to show the empirical impact of concept number on the time it takes to run our method. There’s no theoretical justification, it just was already faster than it needed to be at 500 concepts, so there’s no real reason to see what it does in the “fewer than 500 concepts” domain.
>
> 4. What were the specific reasons for just testing 3B and 4B models? Are there plans to test with other LLMs?
>
> These were the easiest to test on our NVIDIA 4090 GPUs. We have added more tests of some larger models (up to 9B) on H100 GPUs.
>
> 5. Could you elaborate exactly how human oversight would solve the interpretability problem you are trying to address in the paper?
>
> We can clarify this in the text, but essentially what we mean by this is that while this approach qualifies as an “automatic interpretability method” based on the use of a VLM to label concepts, that the need for a human-in-the-loop has not been eliminated due to the fact that the top associated concepts (a small, manageable list) need to be audited for mislabeling. The associations with the outcome are real, the VLM label for what the concept is may just not be.
>
> 6. The work uses OpenCLIP. Did the authors explore other options? Why/why not?
>
> We did not, mostly due to space constraints. We found empirically interesting results with this model, but can definitely test other VLMs as well.

---

### Official Review · Reviewer_8v6i · 2025-11-03

**Soundness:** 2
**Presentation:** 3
**Contribution:** 2
**Rating:** 2
**Confidence:** 3

**Summary:**

This paper introduces Visual Concept Ranking (VCR), an algorithm for auditing large multimodal models by identifying which visual concepts causally influence their outputs. The method learns concept activation vectors (CAVs) by mapping LMM activations to concept scores from an external VLM (like OpenCLIP), operating without expert labels. VCR then ranks these concepts based on the gradient of the LMM's log-probability output with respect to each CAV. It is tested on OpenFlamingo-3B-Instruct and OpenFlamingo-4B on two medical datasets, with better performance than correlation-based concept selection and R^2-based concapt ranking

**Strengths:**

- The paper provides a critical extension of concept-based interpretability to LMMs
- VCR is "automatic" and does not require expensive, expert-annotated concept datasets
- The visualizations are clear and informative, making the audience understand the concepts quickly

**Weaknesses:**

- One major shortcoming is that the interpretability audit is only as reliable as the concept labels provided by the external VLM. The VLMs might have spurious correlation (like the purple ink marking example in line418-431), implicit bias, or lack the nuance for specialized domains. It's unclear how to mitigate this potential risk, especially given the main application is in safety-critical areas.
- The method relies on a predefined set of textual concepts and images. It's unclear how to select the set of text and images, and the effect of size and domain relevance.
- The title ("Automatic Visual Concept Rankings for Large Multimodal Models") suggests a general-purpose method, but all experiments are confined to the medical domain. While the authors suggest it could be applied to general data using vocabularies like Google's Trillion Word Corpus, no such experiments are provided. It is unclear how well this method performs on more abstract or general-domain tasks without this validation.

**Questions:**

- Could the authors elaborate on the novelty of VCR compared to LG-CAV? The CAV-learning pipeline seems to be a direct application of LG-CAV (without the three additional modules).
- What is the exact definition of activation in step 2 (l118-126)?
- As general VLM might lack the nuance for specialized domains, how would VCR's findings change if using a VLM trained in medical data?
- The experiments would be significantly enhanced if more models (such as llava or qwen-vl) can be included.
- Could the authors add visual comparison of activating images of concepts for VCR and baselines?

---

> ### Author Response · Authors · 2025-11-30
> **Thank you**
>
> Thank you for your feedback! To answer your questions point-by-point:
>
> 1) Could the authors elaborate on the novelty of VCR compared to LG-CAV? The CAV-learning pipeline seems to be a direct application of LG-CAV (without the three additional modules).
>
> LG-CAV cannot be directly applied to multimodal models due to the differences in output type, so one bit of novelty is in formulating/defining the output tasks so that the scores can be measured. Another piece of novelty is in demonstrating that the method works without necessitating the additional modules.
>
> 2) What is the exact definition of activation in step 2 (l118-126)?
>
> For a given layer in the model, it is the output of that layer (e.g. not the weights)
>
> 3) As general VLM might lack the nuance for specialized domains, how would VCR's findings change if using a VLM trained in medical data?
>
> This is a great suggestion for a future experiment that we did not include in our paper due to length requirements. It would certainly be interesting to test these results.
>
> 4) The experiments would be significantly enhanced if more models (such as llava or qwen-vl) can be included.
>
> Agreed! We have already updated our results to include MedFlamingo, but could test med Gemini or LLaVA as well.
>
> 5) Could the authors add visual comparison of activating images of concepts for VCR and baselines?
>
> As the concepts are defined using the same VLM for baseline methods, the activating images will be identical. (The gradient concept rankings are the only things that will change).

---

### Meta-Review · Area_Chair_KEUE · 2026-01-02

**Summary:**

Reviewers generally agree that the proposed Visual Concept Ranking (VCR) method is useful for interpreting large multimodal models. The gradient-based analysis is a label-free and automatic pipeline, which is relevant to safety-critical medical tasks.

However, reviewers raised several major concerns: (1) heavy reliance on external VLMs for generating concept labels and the risk of inaccurate or biased labeling, (2) the restricted scope limited to medical domains, which contradicts the authors’ claim that VCR is a general-purpose method, (3) limited novelty compared to LG-CAV, (4) weak experimental results with limited baselines, and (5) the need for human inspection, contradicting the authors’ claim that their method is automatic.

**Reviewer Concerns:**

In rebuttal, the authors clarify a few concerns but most concerns remain unresolved. First, the authors attempted to clarify the limited novelty of the proposed method compared to LG-CAV, mentioning a different formulation of the output tasks but the specifics were not provided. Also, the authors trivialized the reviewer’s concern that the proposed method was reduced to a simpler version of LG-CAV, simply saying that "their method works without additional modules." Other than this no concern is properly addressed by the authors.

**Reviewer Scores:**

Overall, the original evaluations by reviewers ranged from reject to marginally below acceptance threshold, raising multiple substantive concerns. However, the authors largely responded with cursory few-line statements such as "this could be possible", "no reason to see…," or “space constraints" and did not adequately address the issues. Based on this, no change in rating is expected even if the full discussion period is given.

---

### Decision · Program_Chairs · 2026-01-26

Reject